# DeepDFA: Learning and Integration of Regular Languages with Deep Learning

**Elena Umili**  UMILI@DIAG.UNIROMA1.IT

*Sapienza University of Rome*

**Editors:** Leilani H. Gilpin, Eleonora Giunchiglia, Pascal Hitzler, and Emile van Krieken

Most Neuro-Symbolic (NeSy) systems in the current literature are not designed to handle sequential tasks—scenarios where logical rules unfold over time and are best represented through formalisms such as Regular Expressions, Deterministic Finite Automata (DFAs), or Linear Temporal Logic over finite traces (LTLf). Nonetheless, temporal rules are critical across a wide range of domains—including robotics, healthcare, and business process management —to support sequential decision-making and reasoning tasks such as planning, reinforcement learning, and reactive synthesis.

To address this gap, we propose DeepDFA, a general framework for integrating temporal logical knowledge into neural systems, introduced across our recent works Umili and Capobianco (2024); Umili et al. (2024a,b). DeepDFA is a continuous and differentiable logic layer capable of representing temporal rules expressed as DFAs or Moore Machines. Conceptually, it acts as a hybrid between a Recurrent Neural Network (RNN) and a symbolic automaton. Built upon the theory of Probabilistic Finite Automata (PFA), DeepDFA allows temporal logic to be encoded as neural components that are both trainable and compatible with gradient-based optimization. This enables two main capabilities: (i) **Temporal knowledge injection**, where symbolic knowledge is embedded as fixed parameters (as explored in Umili et al. (2024a) and Umili et al. (2024b)), and (ii) **Temporal rule learning**, where the automaton is trained from data (investigated in Umili and Capobianco (2024)).

Regarding temporal knowledge injection, we explored two key settings:

1. **Non-Markovian reinforcement learning in subsymbolic environments**, where decision-making depends on both perception and temporal context.

2. **Symbolic sequence generation**, using deep autoregressive models informed by background temporal knowledge.

In the RL setting, our ECAI 2024 paper Umili et al. (2024a) demonstrates that equipping an RL agent with a purely symbolic representation of its task—encoded via DeepDFA and not grounded a priori in perceptual data—substantially improves both performance and sample efficiency.

In the generative setting, we show that injecting LTLf background knowledge into autoregressive models improves both the compliance of generated sequences with the desired temporal properties and their similarity to ground-truth sequences. These findings are part of an ongoing line of research, with preliminary results presented in Umili et al. (2024b).

As for temporal rule learning, we find that employing DeepDFA offers several advantages: (i) it alleviates **scalability** issues of classical logic-based automata induction algorithms, offering a faster and more memory-efficient alternative; (ii) it is more **accurate** and

less resource-intensive than other NeSy methods and state-of-the-art RNN-to-DFA extraction approaches; (iii) it is **robust to noisy** or mislabeled **training data**; (iv) it better maintains performance under increasing **uncertainty in the input symbol grounding**, in case of probabilistically grounded data sequences. These results are detailed in our conference paper Umili and Capobianco (2024).

In summary, DeepDFA provides a versatile and powerful framework for bridging temporal symbolic reasoning with neural architectures. It supports both the injection of prior temporal knowledge and the learning of temporal rules from data, enabling seamless integration of automata-based logic into deep learning pipelines. These capabilities have been shown to advance the state of the art across multiple domains, including non-Markovian reinforcement learning, autoregressive sequence generation, and automata induction.

## Acknowledgments

This work has been partially supported by PNRR MUR project PE0000013-FAIR.

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
