# OpenReview forum: "DeepDFA: Learning and Integration of Regular Languages with Deep Learning"
_nesyconf.org/NeSy/2025/Conference_Phase_2 — NeSy 2025 - Phase 2 Poster_

### Official Review · Reviewer_Mn5q · 2025-07-05
**Review: DeepDFA: Learning and Integration of Regular Languages with Deep Learning**

**Rating:** 9
**Confidence:** 4

**Review:**

This extended abstract presents DeepDFA, a neuro-symbolic framework that integrates temporal logical knowledge into neural networks. By leveraging a differentiable logic layer based on Probabilistic Finite Automata theory, DeepDFA bridges symbolic temporal reasoning with deep learning, enabling both the injection of prior temporal knowledge and the learning of temporal rules from data. The approach is demonstrated across multiple domains, including non-Markovian reinforcement learning and symbolic sequence generation, showing improvements in performance, efficiency, and robustness.

Motivations, main architecture, and contributions of DeepDFA are clearly stated in this extended abstract. It would be interesting to include potential limitations to have a more balanced view.

**Anonymity:**

Remain anonymous

---

### Official Review · Reviewer_oyDn · 2025-07-05
**Contribution not described**

**Rating:** 4
**Confidence:** 4

**Review:**

In the abstract under review, the author gives an overview of integrating temporal rules into machine learning. Here, they introduce DeepDFA where the rules are encoded as finite automation and are integrated as layer into a recurrent neural network. DeepDFA has been published already in other papers by the author.

Strengths
------------
* Integration of rules, especially temporal rules, is an underexplored in NeSy and informed machine learning

Weaknesses
-----------------
* It is not described, what the paper/abstract contributes that goes beyon what has already been published by the author.

**Anonymity:**

Disclose identity

---

### Official Review · Reviewer_barb · 2025-07-08
**Very interesting, but maybe does not cross the requirement threshold**

**Rating:** 6
**Confidence:** 2

**Review:**

The paper describes DeepDFA --- a NeSy approach for dealing with temporal data. Although the work is quite relevant to NeSy, it may not strictly qualify the threshold set by the organizers regarding being published in "a top-tier conference/journal (e.g., NeurIPS, ICML, ICLR, IJCAI, JMLR, JAIR, etc.)". However, I am cursorily aware of DeepDFA as a body of work, and I think its high relevance to NeSy supersedes the minor differences between conference venues based on subjective ratings.

I lean towards acceptance.

**Anonymity:**

Remain anonymous